# MCUNetV2: Memory-Efficient Patch-based Inference for Tiny Deep Learning

**Ji Lin**[1]    **Wei-Ming Chen**[1]    **Han Cai**[1]    **Chuang Gan**[2]    **Song Han**[1]
[1]MIT    [2]MIT-IBM Watson AI Lab
https://mcunet.mit.edu

## Abstract

Tiny deep learning on microcontroller units (MCUs) is challenging due to the limited memory size. We find that the memory bottleneck is due to the *imbalanced memory distribution* in convolutional neural network (CNN) designs: the first several blocks have an order of magnitude larger memory usage than the rest of the network. To alleviate this issue, we propose a generic *patch-by-patch* inference scheduling, which operates only on a small spatial region of the feature map and significantly cuts down the peak memory. However, naive implementation brings overlapping patches and computation overhead. We further propose *receptive field redistribution* to shift the receptive field and FLOPs to the later stage and reduce the computation overhead. Manually redistributing the receptive field is difficult. We automate the process with neural architecture search to jointly optimize the neural architecture and inference scheduling, leading to MCUNetV2. Patch-based inference effectively reduces the peak memory usage of existing networks by 4-8×. Co-designed with neural networks, MCUNetV2 sets a record ImageNet accuracy on MCU (71.8%), and achieves >90% accuracy on the visual wake words dataset under only 32kB SRAM. MCUNetV2 also unblocks object detection on tiny devices, achieving 16.9% higher mAP on Pascal VOC compared to the state-of-the-art result. Our study largely addressed the memory bottleneck in tinyML and paved the way for various vision applications beyond image classification.

## 1   Introduction

IoT devices based on tiny hardware like microcontroller units (MCUs) are ubiquitous nowadays. Deploying deep learning models on such tiny hardware will enable us to democratize artificial intelligence. However, tiny deep learning is fundamentally different from mobile deep learning due to the tight memory budget [27]: a common MCU usually has an SRAM smaller than 512kB, which is too small for deploying most off-the-shelf deep learning networks. Even for more powerful hardware like Raspberry Pi 4, fitting inference into the L2 cache (1MB) can significantly improve energy efficiency. These pose new challenges to efficient AI inference with a small peak memory usage.

Existing work employs pruning [16, 18, 19, 32, 31], quantization [16, 54, 53, 37, 46, 9, 39], and neural architecture search [6, 44, 47, 5] for efficient deep learning. However, these methods focus on reducing the number of parameters and FLOPs, but not the memory bottleneck. The tight memory budget limits the feature map/activation size, restricting us to use a small model capacity or a small input image size. Actually, the input resolutions used in existing tinyML work are usually small ($< 224^2$) [27], which might be acceptable for image classification (*e.g.*, ImageNet [11], VWW [10]), but not for dense prediction tasks like objection detection: as in Figure 2, the performance of object detection degrades much faster with input resolution than image classification. Such a restriction hinders the application of tiny deep learning on many real-life tasks (*e.g.*, person detection).

35th Conference on Neural Information Processing Systems (NeurIPS 2021).

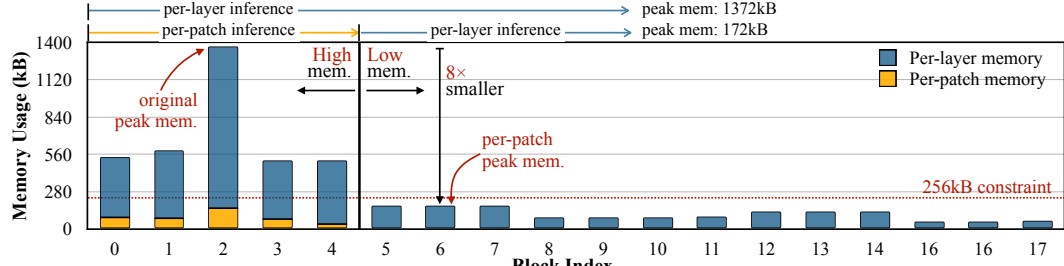

**Figure 1.** MobileNetV2 [41] has a very *imbalanced memory usage distribution*. The peak memory is determined by the first 5 blocks with high peak memory, while the later blocks all share a small memory usage. By using per-patch inference (4 × 4 patches), we are able to significantly reduce the memory usage of the first 5 blocks, and reduce the overall peak memory by 8×, fitting MCUs with a 256kB memory budget. Notice that the model architecture and accuracy are not changed for the two settings. The memory usage is measured in int8.

We perform an in-depth analysis on memory usage of each layer in efficient network designs and find that they have a very *imbalanced activation memory distribution*. Take MobileNetV2 [41] as an example, as shown in Figure 1, only the first 5 blocks have a high peak memory (>450kB), becoming the memory bottleneck of the entire network. The remaining 13 blocks have a low memory usage, which can easily fit a 256kB MCU. The peak memory of the initial memory-intensive stage is 8× higher than the rest of the network. Such a memory pattern leaves a huge room for optimization: if we can find a way to "bypass" the memory-intensive stage, we can reduce the overall peak memory by 8×.

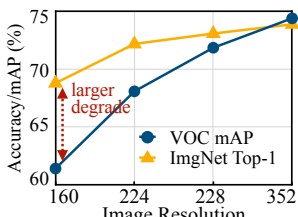

**Figure 2.** Detection is more sensitive to smaller resolutions.

In this paper, we propose MCUNetV2 to address the challenge. We first propose a patch-by-patch execution order for the initial memory-intensive stage of CNNs (Figure 3). Unlike conventional layer-by-layer execution, it operates on a small spatial region of the feature map at a time, instead of the whole activation. Since we only need to store the feature of a small patch, we can significantly cut down the peak memory of the initial stage (blue to yellow in Figure 3), allowing us to fit a larger input resolution. However, the reduced peak memory comes at the price of computation overhead: in order to compute the non-overlapping output patches, the input image patches need to be overlapped (Figure 3(b)), leading to repeated computation. The overhead is positively related to the receptive field of the initial stage: the larger the receptive field, the larger the input patches, which leads to more overlapping. We further propose *receptive field redistribution* to shift the receptive field and workload to the later stage of the network. This reduces the patch size as well as the computation overhead caused by overlapping, without hurting the performance of the network. Finally, patch-based inference brings a larger design space for the neural network, giving us more freedom trading-off input resolution, model size, *etc*. We also need to minimize the computation overhead under patch-based execution. To explore such a large and entangled space, we propose to jointly design the optimal deep model and its inference schedule with neural architecture search given a specific dataset and hardware.

Patch-based inference effectively reduces the peak memory usage of existing networks by 4-8× (Figure 5). The results are further improved when co-designing neural architecture with inference scheduling. On ImageNet [11], we achieve a record accuracy of 71.8% on MCU (Table 2); on visual wake words dataset [10], we are able to achieve >90% accuracy under only 32kB SRAM, which is 4.0× smaller compared to MCUNetV1 [27], greatly lowering the boundary of tiny deep learning (Figure 7). MCUNetV2 further unlocks the possibility to perform dense prediction tasks on MCUs (*e.g.*, object detection), which was not practical due to the limited input resolution. We are able to achieve 64.6% mAP under 256kB SRAM constraints and 68.3% under 512kB, which is 16.9% higher compared to the existing state-of-the-art solution, making object detection applicable on a tiny ARM Cortex-M4 MCU. Our contributions can be summarized as follows:

- We systematically analyze the memory usage pattern of efficient CNN designs and find that they suffer from a *imbalanced memory distribution*, leaving a huge room for optimization.
- We propose a patch-based inference scheduling to significantly reduce the peak memory required for running CNN models, together with receptive field redistribution to minimize the computation overhead.

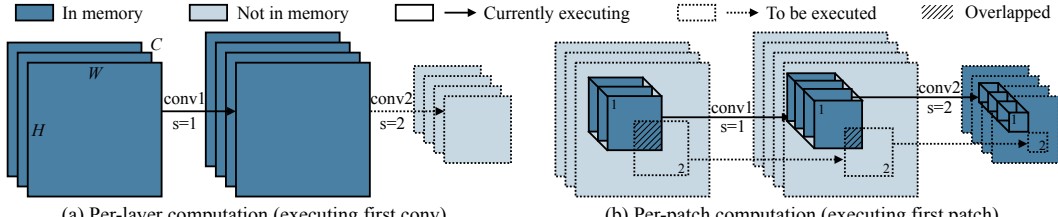

| (a) Per-layer computation (executing first conv) | (b) Per-patch computation (executing first patch) |

**Figure 3.** Per-patch inference can significantly reduce the peak memory required to execute a sequence of convolutional layers. We study two convolutional layers (stride 1 and 2). Under per-layer computation (a), the first convolution has a large input/output activation size, dominating the peak memory requirement. With per-patch computation (b), we allocate the buffer to host the final output activation, and compute the results *patch-by-patch*. We only need to store the activation from *one patch* but not the entire feature map, reducing the peak memory (the first input is the image, which can be partially decoded from a compressed format like JPEG).

- With the joint design of network architecture and inference scheduling, we achieve a record performance for tiny image classification and objection detection on MCUs. Our work largely addressed the memory bottleneck for tinyML, paving the way for various vision applications.

## 2    Understanding the Memory Bottleneck of Tiny Deep Learning

We systematically analyze the memory bottleneck of CNN models.

**Imbalanced memory distribution.**    As an example, we provide the per-block peak memory usage of MobileNetV2 [41] in Figure 1. The profiling is done in int8 (details in Section 4). We can observe a clear pattern of *imbalanced memory usage distribution*. The first 5 blocks have large peak memory, exceeding the memory constraints of MCUs, while the remaining 13 blocks easily fit 256kB memory constraints. The third block has $8\times$ larger memory usage than the rest of the network, becoming the memory bottleneck. We also inspect other efficient network designs and find the phenomenon quite common across different CNN backbones, even for models specialized for memory-limited microcontrollers [27]. The detailed statistics are provided in the supplementary.

We find that this situation applies to most single-branch or residual CNN designs due to the hierarchical structure*: after each stage, the image resolution is down-sampled by half, leading to $4\times$ fewer pixels, while the channel number increases only by $2\times$ [42, 17, 22] or by an even smaller ratio [41, 21, 45], resulting in a decreasing activation size. Therefore, the memory bottleneck tends to appear at the early stage of the network, after which the peak memory usage is much smaller.

**Challenges and opportunities.**    The imbalanced memory distribution significantly limits the model capacity and input resolution executable on MCUs. In order to accommodate the initial memory-intensive stage, the whole network needs to be scaled down even though the majority of the network already has a small memory usage. It also makes resolution-sensitive tasks (*e.g.*, object detection) difficult, as a high-resolution input will lead to large initial peak memory. Consider the first convolutional layer in MobileNetV2 [41] with input channels 3, output channels 32, and stride 2, running it on an image of resolution $224 \times 224$ requires a memory of $3 \times 224^2 + 32 \times 112^2 = 539$kB even when quantized in int8, which cannot be fitted into microcontrollers. On the other hand, if we can find a way to "bypass" the initial memory-intensive stage, we can greatly reduce the peak memory of the whole network, leaving us a large room for optimization.

## 3    MCUNetV2: Memory-Efficient Patch-based Inference

### 3.1    Breaking the Memory Bottleneck with Patch-based Inference

We propose to break the memory bottleneck of the initial layers with *patch-based inference* (Figure 3). Existing deep learning inference frameworks (*e.g.*, TensorFlow Lite Micro [1], TinyEngine [27], microTVM [8], *etc.*) use a *layer-by-layer* execution. For each convolutional layer, the inference library first allocates the input and output activation buffer in SRAM, and releases the input buffer after the *whole* layer computation is finished. Such an implementation makes inference optimization easy (*e.g.*, im2col, tiling, *etc.*), but the SRAM has to hold the entire input and output activation for

---

*some CNN designs have highly complicated branching structure (*e.g.*, NASNet [56]), but they are generally less efficient for inference [34, 44, 6]; thus not widely used for edge computing.

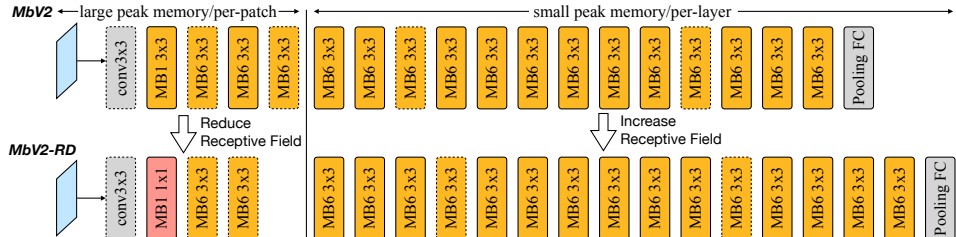

**Figure 4.** The redistributed MobileNetV2 (MbV2-RD) has reduced receptive field for the per-patch inference stage and increased receptive field for the per-layer stage. The two networks have the same level of performance, but MbV2-RD has a smaller overhead under patch-based inference. The mobile inverted block is denoted as `MB{expansion ratio} {kernel size}`. The dashed border means stride=2.

**Table 1.** Per-patch inference reduces the peak memory by $8\times$ for MobileNetV2 [41] (1372kB to 172kB), but it increases the overall computation by 10% due to patch overlapping. We futher propose receptive field redistribution (MbV2-RD) which reduces the overall overhead to only 3% without hurting performance.

| Model | Patch Size | Comp. overhead | | MACs$_{(4\times4\ \text{patches})}$ | | Peak SRAM | | ImgNet Top-1 | VOC mAP |
|---|---|---|---|---|---|---|---|---|---|
| | | patch-stage | overall | patch-stage | overall | per-layer | per-patch | | |
| MbV2 [41] | $75^2$ | +42% | +10% | 130M | 330M | 1372kB | 172kB ($\mathbf{8\times\downarrow}$) | 72.2% | 75.4% |
| MbV2-RD | $63^2$ | +18% | +3% | 73M | 301M | 1372kB | 172kB ($\mathbf{8\times\downarrow}$) | 72.1% | 75.7% |

each layer, which is prohibitively large for the initial stage of the CNN. Our patch-based inference runs the initial memory-intensive stage in a *patch-by-patch* manner. For each time, we only run the model on a small spatial region ($>10\times$ smaller than the whole area), which effectively cuts down the peak memory usage. After this stage is finished, the rest of the network with a small peak memory is executed in a normal layer-by-layer manner (upper notations in Figure 1).

We show an example of two convolutional layers (with stride 1 and 2) in Figure 3. For conventional per-layer computation, the first convolutional layer has large input and output activation size, leading to a high peak memory. With spatial partial computation, we allocate the buffer for the final output and compute its values *patch-by-patch*. In this manner, we only need to store the activation from *one patch* instead of the *whole feature map*. Note that the first activation is the input image, which can be partially decoded from a compressed format like JPEG and does not require full storage.

**Computation overhead.** The significant memory saving comes at the cost of computation overhead. To maintain the same output results as per-layer inference, the non-overlapping output patches correspond to overlapping patches in the input image (the shadow area in Figure 3(b)). This is because convolutional filters with kernel size >1 contribute to increasing receptive fields. The bordering pixel on the output patches is dependent on the inputs from neighboring patches. Such repeated computation can increase the overall network computation by 10-17% even under optimal hyper-parameter choice (Figure 5), which is undesirable for low-power edge devices.

## 3.2 Reducing Computation Overhead by Redistributing the Receptive Field

The computation overhead is related to the receptive field of the patch-based initial stage. Consider the output of the patch-based stage, the larger receptive field it has on the input image, the larger resolution for each patch, leading to a larger overlapping area and repeated computation (see Section 4.4 for quantitative analysis). For MobileNetV2, if we only consider down-sampling, each input patch has a side length of $224/4 = 56$. But when considering the increased receptive field, each input patch has to use a shape of $75 \times 75$, leading to a large overlapping area.

We propose to *redistribute* the receptive field (RF) of the CNN to reduce computation overhead. The basic idea is: *(1) reduce the receptive field of the patch-based initial stage; (2) increase the receptive field of the later stage*. Reducing RF for the initial stage helps to reduce the size of each input patch and repeated computation. However, some tasks may have degraded performance if the overall RF is smaller (*e.g.*, detecting large objects). Therefore, we further increase the RF of the later stage to compensate for the performance loss.

We take MobileNetV2 as a study case and modify its architecture. The comparison is shown in Figure 4. We used smaller kernels and fewer blocks in the per-patch inference stage, and increased

the number of blocks in the later per-layer inference stage. The process needs manual tuning and varies case-by-case. We will later discuss how we *automate* the process with NAS. We compare the performance of the two architectures in Table 1. Per-patch inference reduces the peak SRAM by $8\times$ for all cases, but the original MobileNetV2 design has 42% computation overhead for the patch-based stage and 10% for the overall network. After redistributing the receptive field ("MbV2-RD"), we can reduce the input patch size from 75 to 63, while maintaining the same level of performance in image classification and object detection. After redistribution, the computation overhead is only 3%, which is negligible considering the benefit in memory reduction.

### 3.3 Joint Neural Architecture and Inference Scheduling Search

Redistributing the receptive field allows us to enjoy the benefit of memory reduction at minimal computation/latency overhead, but the strategy varies case-by-case for different backbones. The reduced peak memory also allows larger freedom when designing the backbone architecture (*e.g.*, using a larger input resolution). To explore such a large design space, we propose to jointly optimize the *neural architecture* and the *inference scheduling* in an automated manner. Given a certain dataset and hardware constraints (SRAM limit, Flash limit, latency limit, *etc*.), our goal is to achieve the highest accuracy while satisfying all the constraints. We have the following knobs to optimize:

**Backbone optimization.** We follow [27] to use a MnasNet-alike search space [44, 6] for NAS, so that we can have a fair comparison. The space includes different kernel sizes for each inverted residual block $k_{[\ ]}$ (3/5/7), different expansion ratios $e_{[\ ]}$ (3/4/6), and a different number of blocks for each stage $d_{[\ ]}$ (2/3/4). More recent search space designs like MobileNetV3 [21] have better accuracy-computation trade-off, but are hard to quantize due to Swish activation function [36], making deployment on MCU difficult. As shown in [27], the search space configuration (*i.e.*, the global width multiplier $w$ and input resolution $r$) is crucial to the final NAS performance. We argue that the best search space configuration is not only *hardware-aware* but also *task-aware*: for example, some tasks may prefer a higher resolution over a larger model size, and vice versa. Therefore, we also put $r$ and $w$ into the search space. We further extend $w$ to support per-block width scaling $w_{[\ ]}$. Including $w_{[\ ]}$ (0.5/0.75/1.0) and $r$ (96-256) expands the search space scalability, allowing us to fit different MCU models and tight resource budgets (ablation study provided in the supplementary).

**Inference scheduling optimization.** Given a model and hardware constraints, we will find the best inference scheduling. Our inference engine is developed based on TinyEngine [27] to further patch-based inference. Apart from the optimization knobs in TinyEngine, we also need to determine the patches number $p$ and the number of blocks $n$ to perform patch-based inference, so that the inference satisfies the SRAM constraints. According to Section 4.4, a smaller $p$ and $n$ lead to a smaller computation overhead and faster inference. But it varies case-by-case, so we jointly optimize it with the architecture.

**Joint search.** We need to co-design the backbone optimization and inference scheduling. For example, given the same constraints, we can choose to use a smaller model that fits per-layer execution ($p = 1$, no computation overhead), or a larger model and per-patch inference ($p > 1$, with a small computation overhead). Therefore, we put both sides in the same loop and use evolutionary search to find the best set of $(k_{[\ ]}, e_{[\ ]}, d_{[\ ]}, w_{[\ ]}, r, p, n)$ satisfying constraints. Specifically, we randomly sample neural networks from the super network search space; for each sampled network, we enumerate all the $p$ and $n$ choices (optimized together with other knobs in TinyEngine) and find the satisfying combinations. We then report the best $(p, n)$ pair with minimal computation/latency, and use the statistics to supervise architecture search. We provide the details and pseudo code in supplementary.

## 4 Experiments

**Memory profiling.** The memory usage is dependent on the inference framework implementation [27]. To ease the comparison, we study two profiling settings:

(1) We first study *analytic profiling*, which is only related to the model architecture but not the inference framework. Following [10, 40], the memory required for a layer is the sum of input and output activation (since weights can be partially fetched from Flash); for networks with multi-branches (*e.g.*, residual connection), we consider the sum of memory required for all branches at the same time (if the same input is shared by multiple branches, it will only be counted once).

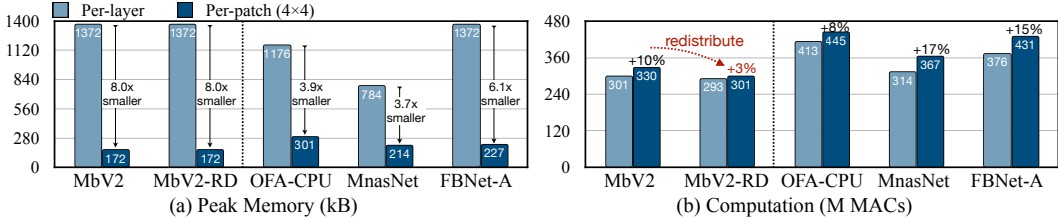

**Figure 5. Analytical profiling**: patch-based inference significantly reduces the inference peak memory by **3.7-8.0×** at a small computation overhead of 8-17%. The memory reduction and computation overhead are related to the network design. For MobileNetV2, we can reduce the computation overhead from 10% to 3% by redistributing the receptive field. All networks take an input resolution of $224^2$ and $4 \times 4$ patches.

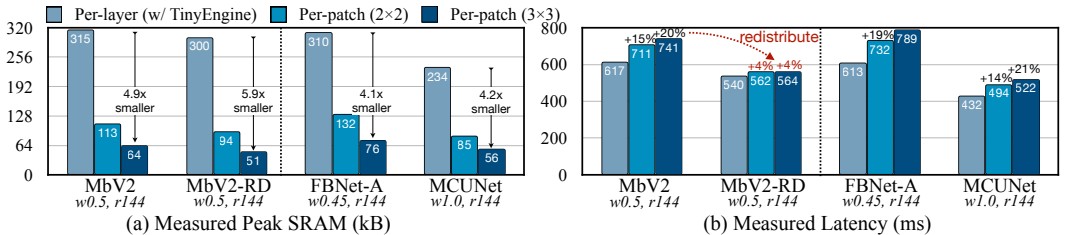

**Figure 6. On-device measurement**: patch-based inference reduce the *measured* peak SRAM usage by **4-6×** when running on MCUs. The latency overhead could be large for some networks, but we can reduce it to 4% with proper architecture design (MbV2-RD). We scale down width $w$ and resolution to fit MCU memory.

(2) We also study *on-device profiling* to report the *measured* SRAM and Flash usage when executing the deep model on MCU. The number is usually larger than the analytic results since we need to account for temporary buffers storing partial weights, Im2Col buffer, *etc*.

**Datasets.** We analyze the advantage of our method on image classification datasets: ImageNet [11] as the standard benchmark, and Visual Wake Words [10] to reflect TinyML applications. We further validate our method on object detection datasets: Pascal VOC [13] and WIDER FACE [48] to show our advantage: be able to fit larger resolution on the MCU.

**Training&deployment.** We follow [27] for super network training and evolutionary search, detailed in the supplementary. Models are quantized to `int8` for deployment. We extend TinyEngine [27] to support patch-based inference, and benchmark the models on 3 MCU models with different hardware resources: STM32F412 (Cortex-M4, 256kB SRAM/1MB Flash), STM32F746 (Cortex-M7, 320kB SRAM/1MB Flash), STM32H743 (Cortex-M7, 512kB SRAM/2MB Flash).

### 4.1 Reducing Peak Memory of Existing Networks

We first analyze how patch-based inference can significantly reduce the peak memory for model inference, both in analytic profiling and on-device profiling.

**Analytic profiling.** We study several widely used deep network backbones designed for edge inference in Figure 5: MobileNetV2 [41] (MbV2), redistributed MobileNetV2 (MbV2-RD), Once-For-All CPU (OFA-CPU) [5], MnasNet [44], and FBNet-A [47]. All the networks use an input resolution of $224 \times 224$; for patch-based inference, we used $4 \times 4$ patches. The memory is profiled in `int8`. Per-patch inference significantly reduces the peak memory by **3.7-8.0×**, while only incurring **8-17%** of computation overhead. For MobileNetV2, we can reduce the computation overhead from 10% to 3% by redistributing the receptive field without hurting accuracy (Table 1). The memory saving and computation reduction are related to the network architecture. Some models like MnasNet have a larger overhead since it uses large kernel sizes in the initial stage, which increases receptive fields. It shows the necessity to *co-design* the network architecture with the inference engine.

**On-device measurement.** We further profile existing networks running on STM32F746 MCU. We measure the SRAM usage of the network with per-layer and per-patch ($2 \times 2$ or $3 \times 3$ patches) inference. Due to the memory limit of MCU (320kB SRAM, 1MB Flash), we have to scale down the width multiplier $w$ and input resolution $r$. As in Figure 6, per-patch based inference reduces the measured peak SRAM by 4-6×. Some models may have a large latency overhead, since the initial

**Table 2.** MCUNetV2 significantly improves the ImageNet accuracy on microcontrollers, outperforming the state-of-the-arts by **4.6%** under 256kB SRAM and **3.3%** under 512kB. Lower or mixed precisions (marked gray) are orthogonal techniques, which we leave for future work. Out-of-memory (OOM) results are ~~struck out~~.

| Model / Library | Quant. | MACs | SRAM | Flash | Top-1 | Top-5 |
|---|---|---|---|---|---|---|
| *STM32F412 (256kB SRAM, 1MB Flash)* | | | | | | |
| MbV2 0.35× ($r$=144) [41] / TinyEngine [27] | int8 | 24M | ~~308kB~~ | 862kB | 49.0% | 73.8% |
| Proxyless 0.3× ($r$=176) [6] / TinyEngine [27] | int8 | 38M | ~~292kB~~ | 892kB | 56.2% | 79.7% |
| MbV1 0.5× ($r$=192) [22] / Rusci *et al.* [39] | mixed | 110M | <256kB | <1MB | 60.2% | |
| MCUNet (TinyNAS / TinyEngine) [27] | int8 | 68M | 238kB | 1007kB | 60.3% | - |
| MCUNet (TinyNAS / TinyEngine) [27] | int4 | 134M | 233kB | 1008kB | 62.0% | - |
| MCUNetV2-M4 | int8 | 119M | 196kB | 1010kB | **64.9%** | **86.2%** |
| *STM32H743 (512kB SRAM, 2MB Flash)* | | | | | | |
| MbV1 0.75× ($r$=224) [22] / Rusci *et al.* [39] | mixed | 317M | <512kB | <2MB | 68.0% | |
| MCUNet (TinyNAS / TinyEngine) [27] | int8 | 126M | 452kB | 2014kB | 68.5% | - |
| MCUNet (TinyNAS / TinyEngine) [27] | int4 | 474M | 498kB | 2000kB | 70.7% | - |
| MCUNetV2-H7 | int8 | 256M | 465kB | 2032kB | **71.8%** | **90.7%** |

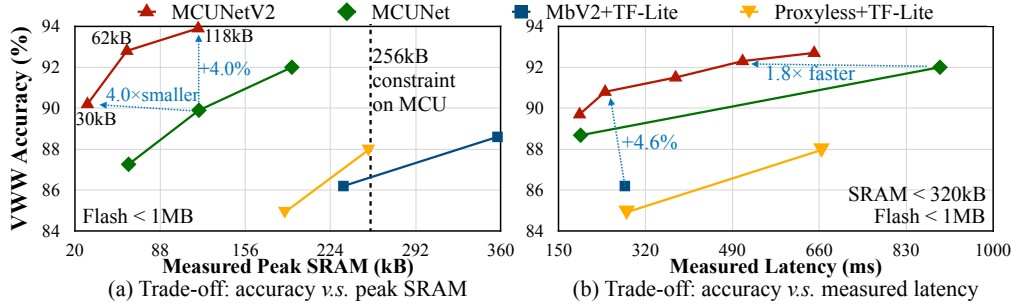

(a) Trade-off: accuracy *v.s.* peak SRAM      (b) Trade-off: accuracy *v.s.* measured latency

**Figure 7. Left**: MCUNetV2 has better visual wake word (VWW) accuracy *vs.* peak SRAM trade-off. Compared to MCUNet [27], MCUNetV2 achieves better accuracy at 4.0× smaller peak memory. It achieves >90% accuracy under <32kB memory, facilitating deployment on extremely small hardware. **Right**: patch-based method expands the search space that can fit the MCU, allowing better accuracy *vs.* latency trade-off.

stage has worse hardware utilization. But with a proper architecture design (MbV2-RD), we can reduce the latency overhead to 4%, which is negligible compared to the memory reduction benefit.

### 4.2 MCUNetV2 for Tiny Image Classification

With joint optimization of neural architecture and inference scheduling, MCUNetV2 significantly pushes the state-of-the-art results for MCU-based tiny image classification.

**Pushing the ImageNet record on MCUs.** We compared MCUNetV2 with existing state-of-the-art solutions on ImageNet classification under two hardware settings: 256kB SRAM/1MB Flash and 512kB SRAM/2MB Flash. The former represents a widely used Cortex-M4 microcontroller; the latter corresponds to a higher-end Cortex-M7. The goal is to achieve the highest ImageNet accuracy on resource-constrained MCUs (Table 2). MCUNetV2 significantly improves the ImageNet accuracy of tiny deep learning on microcontrollers. Under 256kB SRAM/1MB Flash, MCUNetV2 outperforms the state-of-the-art method [27] by 4.6% at 18% lower peak SRAM. Under 512kB SRAM/2MB Flash, MCUNetV2 achieves a new *record* ImageNet accuracy of 71.8% on commercial microcontrollers, which is 3.3% compared to the best solution under the same quantization policy. Lower-bit (`int4`) or mixed-precision quantization can further improve the accuracy (marked in gray in the table). We believe that we can further improve the accuracy of MCUNetV2 with a better quantization policy, which we leave to future work.

**Visual Wake Words under 32kB SRAM.** Visual wake word (VWW) reflects the low-energy application of tinyML. MCUNetV2 allows us to run a VWW model with a modest memory requirement. As in Figure 7, MCUNetV2 outperforms state-of-the-art method [27] for both accuracy *vs.*

**Table 3.** MCUNetV2 significantly improves Pascal VOC [13] object detection on MCU by allowing a higher input resolution. Under STM32H743 MCU constraints, MCUNetV2-H7 improves the mAP by 16.9% compared to [27], achieving a record performance on MCU. It can also scale down to cheaper MCU STM32F412 with only 256kB SRAM while still improving mAP by 13.2% at 1.9× smaller peak SRAM and a similar computation.

| MCU Model | Constraint | Model | #Param | MACs | peak SRAM | VOC mAP | Gain |
|---|---|---|---|---|---|---|---|
| H743 (~$7) | SRAM <512kB | MbV2+CMSIS [27] | 0.87M | 34M | ~~519kB~~ | 31.6% | - |
| | | MCUNet [27] | 1.20M | 168M | 466kB | 51.4% | 0% |
| | | MCUNetV2-H7 | 0.67M | 343M | 438kB | **68.3%** | +16.9% |
| F412 (~$4) | <256kB | MCUNetV2-M4 | 0.47M | 172M | **247kB** | 64.6% | +13.2% |

**Table 4.** MCUNetV2 outperforms existing methods for memory-efficient face detection on WIDER FACE [48] dataset. Compared to RNNPool-Face-C [40], MCUNetV2-L can achieve similar mAP at **3.4×** smaller peak SRAM and **1.6×** smaller computation. The model statistics are profiled on 640 × 480 RGB input images following [40].

| Method | MACs ↓ | Peak Memory ↓ (fp32) | mAP ↑ | | | mAP (≤3 faces) ↑ | | |
|---|---|---|---|---|---|---|---|---|
| | | | Easy | Medium | Hard | Easy | Medium | Hard |
| EXTD [49] | 8.49G | 18.8MB (9.9×) | 0.90 | 0.88 | 0.82 | 0.93 | 0.93 | 0.91 |
| LFFD [20] | 9.25G | 18.8MB (9.9×) | 0.91 | 0.88 | 0.77 | 0.83 | 0.83 | 0.82 |
| RNNPool-Face-C [40] | 1.80G | 6.44MB (3.4×) | **0.92** | 0.89 | **0.70** | **0.95** | **0.94** | **0.92** |
| MCUNetV2-L | **1.10G** | **1.89MB** (1.0×) | **0.92** | **0.90** | **0.70** | 0.94 | 0.93 | **0.92** |
| EagleEye [52] | **0.08G** | 1.17MB (1.8×) | 0.74 | 0.70 | 0.44 | 0.79 | 0.78 | 0.75 |
| RNNPool-Face-A [40] | 0.10G | 1.17MB (1.8×) | 0.77 | 0.75 | 0.53 | 0.81 | 0.79 | 0.77 |
| MCUNetV2-S | 0.11G | **672kB** (1.0×) | **0.85** | **0.81** | **0.55** | **0.90** | **0.89** | **0.87** |

peak memory and accuracy *vs.* latency trade-off. We perform neural architecture search under both *per-layer* and *per-patch* inference settings using the same search space and super network for ablation. Compared to per-layer inference, MCUNetV2 can achieve better accuracy using 4.0× smaller memory. Actually, it can achieve >90% accuracy under 32kB SRAM requirement, allowing us to deploy the model on low-end MCUs like STM32F410 costing only $1.6. For the latency-constrained setting, we jointly optimized the model architecture and inference scheduling, where a smaller patch number is used when possible. Per-patch inference also expands the search space, giving us more freedom to find models with better accuracy *vs.* latency trade-off.

### 4.3 MCUNetV2 for Tiny Object Detection

Object detection is sensitive to a smaller input resolution (Figure 2). Current state-of-the-art [27] cannot achieve a decent detection performance on MCUs due to the resolution bottleneck. MCUNetV2 breaks the memory bottleneck for detectors and improves the mAP by double digits.

**MCU-based detection on Pascal VOC.** We show the object detection results on Pascal VOC trained with YOLOv3 [38] on Table 3. We provide MCUNetV2 results for M4 MCU with 256kB SRAM and H7 MCU with 512kB SRAM. On H7 MCU, MCUNetV2-H7 improves the mAP by 16.7% compared to the state-of-the-art method MCUNet [27]. It can also scale down to fit a cheaper commodity Cortex-M4 MCU with only 256kB SRAM, while still improving the mAP by 13.2% at 1.9× smaller peak SRAM. Note that MCUNetV2-M4 shares a similar computation with MCUNet (172M *vs.* 168M) but a much better mAP. This is because the expanded search space from patch-based inference allows us to choose a better configuration of larger input resolution and smaller models.

**Memory-efficient face detection.** We benchmarked MCUNetV2 for memory-efficient face detection on WIDER FACE [48] dataset in Table 4. We report the analytic memory usage of the detector backbone in fp32 following [40]. We train our methods with S3FD face detector [50] following [40] for a fair comparison. We also report mAP on samples with ≤ 3 faces, which is a more realistic setting for tiny devices. MCUNetV2 outperforms existing solutions under different scales. MCUNetV2-L achieves comparable performance at 3.4× smaller peak memory compared to RNNPool-Face-C [27] and 9.9× smaller peak memory compared to LFFD [20]. The computation is also 1.6× and 8.4×

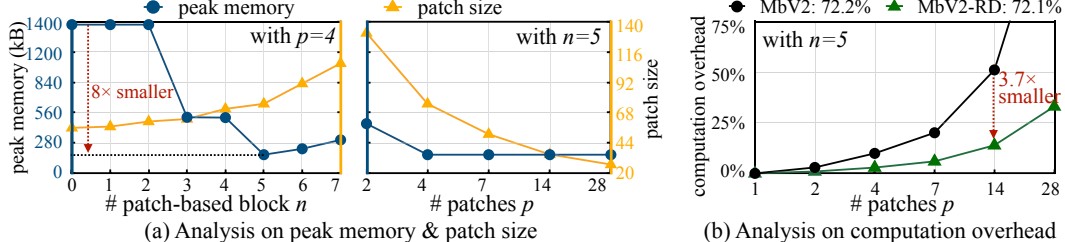

(a) Analysis on peak memory & patch size

(b) Analysis on computation overhead

**Figure 8.** Ablation study on patch-based inference. **Left**: the peak memory generally goes down with more blocks being executed patch-by-patch ($n$) and a larger patch number ($p$). The optimal index for MobileNetV2 is $n^* = 5$, where the feature map is down-sampled by $8\times$. **Right**: splitting the input images into more patches (larger $p$) leads to larger computation overhead. Receptive field redistribution reduces the overhead (MbV2-RD).

**Table 5.** Comparing MCUNetV2 with other memory-saving methods. Non-overlapping patches suffer from a degraded detection performance; RNNPool [40] leads to worse performance and slower training time. MCUNetV2 with redistributed model maintains the accuracy at the same training cost. Degraded items marked in red.

| Model | Inference | Invariant | Peak Mem$_{fp32}$ | Train time | ImgNet Top-1 | VOC mAP |
|---|---|---|---|---|---|---|
| MbV2 [41] | Per-layer | ✓ | 2.29MB | 1.0× | 72.2% | 75.4% |
| Non-overlap | Per-patch | ✗ | **0.19MB** | 1.0× | 71.8% | 73.9% |
| MbV2-RNNPool [40] | Streaming | ✗ | 0.24MB | 3.2× | 70.1% | 71.0% |
| MbV2-RD (ours) | Per-patch | ✓ | **0.19MB** | 1.0× | 72.1% | 75.7% |

smaller. MCUNetV2-S consistently outperforms RNNPool-Face-A [40] and EagleEye [52] at $1.8\times$ smaller peak memory.

### 4.4 Analysis

**Hyper-parameters for patch-based inference.** We study the hyper-parameters used for patch-based inference: the number of blocks to be executed under patch-based inference $n$; the number of patches to split the input image $p$ (splitting the image into $p \times p$ overlapping patches). We analyze MobileNetV2 [41] in Figure 8(a), with a larger $n$, the patch size increases due to the growing receptive field. The peak memory first goes down since the output feature map is smaller then goes up due to larger receptive field overhead. $n = 5$ is optimal. For a larger $p$ (given $n$=5), each patch is smaller, which helps to reduce the peak memory. However, it also leads to more computation overhead due to more spatial overlapping (Figure 8(b)). Receptive field redistribution can reduce the overhead significantly (MbV2-RD). The optimal design is $n^* = 5, p^* = 4$ to reach the minimum peak memory with the smallest overhead. The choice of $p$ and $n$ varies for different networks. Therefore, we use an automated method to jointly optimize with neural architecture (Section 3.3).

**Comparison to other solutions.** We also compare MCUNetV2 with other methods that reduce inference peak memory. The comparison on MobileNetV2 [41] is shown in Table 5. A straightforward way is to split the input image into *non-overlapping* patches ("Non-overlap") for the first several blocks as done in [12]. Such a practice does not incur extra computation, but it breaks the feature propagation between patches and the translational invariance of CNNs. It achieves lower image classification accuracy and significantly degraded object detection mAP (on Pascal VOC) due to the lack of cross-patch communication (a similar phenomenon is observed in [30]). For MobileNetV2 with RNNPool [40], it can reduce the peak memory but leads to inferior ImageNet accuracy and object detection mAP. Its training time is also $3.2\times$ longer[†] due to the complicated data path and the RNN module. On the other hand, MCUNetV2 acts exactly the same as a normal network during training (per-layer forward/backward), while also matching the image classification and objection detection performance. MCUNetV2 can further improve the results with joint neural architecture and inference scheduling search (Section 4.2).

**Dissecting MCUNetV2 architecture.** We visualize one of the MCUNetV2 model architecture on the VWW [10] dataset in Figure 9. We can find the following patterns:

- The kernel size in the per-patch inference stage is small ($1 \times 1$ and $3 \times 3$) to reduce the receptive field and spatial overlapping, thus reducing computation overhead.

---

[†]measured with official PyTorch code (MIT License) using a batch size of 64 on NVIDIA Titan RTX.

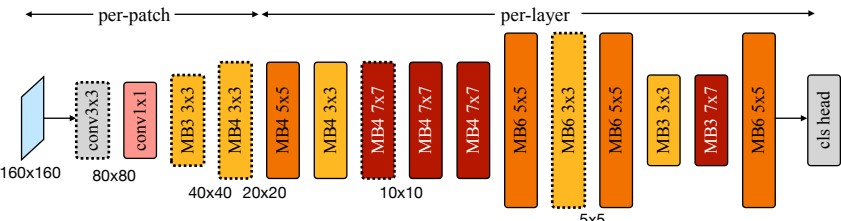

**Figure 9.** An MCUNetV2 architecture on VWW. The color represents the kernel size; the height of each block represents the expansion ratio. The name is `MB{expansion ratio} {kernel size}x{kernel size}`. Blocks with dashed borders have stride=2. `{}x{}` in the bottom denotes the feature map resolution.

- The expansion ratio of the middle stage (early in per-layer stage) is small to further reduce the peak memory; while the expansion ratio for the later stage is large to increase performance.
- Large expansion ratios and large kernel sizes usually do not appear together to reduce the computational cost and latency: if the expansion ratio is large (like 6), the kernel size is small ($3 \times 3$ or $5 \times 5$); if the kernel size is large ($7 \times 7$), the expansion ratio is small (3 or 4).
- The input resolution is larger on resolution-sensitive datasets like VWW compared to MCUNet [27], since per-layer inference cannot fit a large input resolution.

Notice that all the patterns are automatically discovered by the joint neural architecture and inference scheduling search algorithm, without human expertise.

## 5 Related Work

**Tiny deep learning on microcontrollers.** Deploying deep learning models on memory-constrained microcontrollers requires an efficient inference framework and model architecture. Existing deployment frameworks include TensorFlow Lite Micro [1], CMSIS-NN [23], TinyEngine [27], MicroTVM [8], CMix-NN [7], *etc*. However, all of the above frameworks support only *per-layer* inference, which limits the model capacity executable under a small memory and makes higher resolution input impossible.

**Efficient neural network.** For efficient deep learning, people apply pruning [16, 18, 28, 19, 32, 31] and quantization [16, 54, 53, 37, 46, 9, 39, 24] to compress an off-the-shelf deep network, or directly design an efficient network architecture [22, 41, 21, 34, 51]. Neural architecture search (NAS) [55, 56, 29, 6, 44, 47] can design efficient models in an automated way. It has been used to improve tinyML on MCUs [27, 4, 25, 14, 33]. However, most of the NAS methods use the conventional hierarchy CNN backbone design, which leads to an imbalanced memory distribution under per-layer inference (Section 2), restricting the input resolution. Therefore, they are not able to achieve good performance on tasks like object detection without our patch-based inference scheduling.

**Computation scheduling/re-ordering.** The memory requirement to run a deep neural network is related to the implementation. It is possible to reduce the required memory by optimizing the convolution loop-nest [43], reordering the operator executions [26, 2], or temporarily swapping data off SRAM [35]. Computing partial spatial regions across multiple layers can reduce the peak memory [15, 3, 40]. However, system-only optimization leads to either large repeated computation or a highly complicated dataflow. Our work explores joint system and model optimization to reduce the peak memory at a negligible computation overhead while still allowing conventional convolution optimization techniques like im2col, tiling, *etc*.

## 6 Conclusion

In this paper, we propose patch-based inference to reduce the memory usage for tiny deep learning by up to $8\times$, which greatly expands the design space and unlocks powerful vision applications on IoT devices. We jointly optimize the neural architecture and inference scheduling to develop MCUNetV2. MCUNetV2 significantly improves the object detection performance on microcontrollers by 16.9% and achieves a record ImageNet accuracy (71.8%). For the VWW dataset, MCUNetV2 can achieve >90% accuracy under only 32kB SRAM, $4\times$ smaller than existing work. Our study largely addresses the memory bottleneck in tinyML and paves the way for vision applications beyond classification.

## Acknowledgments

We thank MIT-IBM Watson AI Lab, Samsung, Woodside Energy, and NSF CAREER Award #1943349 for supporting this research.

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
