# Supplementary Material
# MCUNetV2: Memory-Efficient Patch-based Inference for Tiny Deep Learning

## Contents

35th Conference on Neural Information Processing Systems (NeurIPS 2021).

# A Flow Chart of Contributions

We provide a flow chart to summarize our contributions in Figure S1.

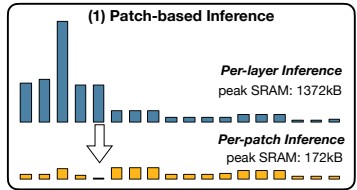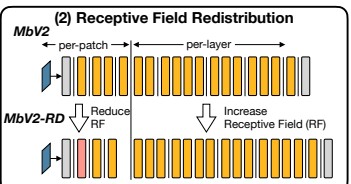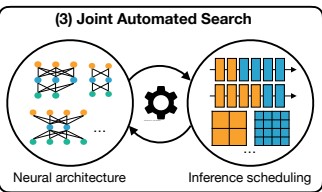

**Figure S1.** Contributions of MCUNetV2: (1) Analyze and find the imbalanced memory distribution; propose a patch-based inference scheduling to reduce the peak memory significantly; (2) Propose redistributing receptive fields to reduce the overhead from overlapping patches; (3) Jointly optimize the neural architecture and inference scheduling in the same loop.

# B Experimental Details

**Search space.** We used a MnasNet-alike search space [12, 9, 1] for neural architecture search. The search space consists with the following knobs:

- Kernel size for each separable convolution block $k_{[\,]}$, choosing from $\{3, 5, 7\}$.
- Expansion ratio for each inverted residual block $e_{[\,]}$, choosing from $\{3, 4, 6\}$.
- Number of blocks for each stage $d_{[\,]}$, choosing from $\{2, 3, 4\}$.
- Width multiplier for each block $w_{[\,]}$, choosing from $\{0.5, 0.75, 1.0\}$.
- Input image resolution $r$, choosing from $\{96, 128, 160, 192, 224, 256\}$.

For the inference scheduling, apart from the optimization knobs inherited from TinyEngine [9], we also include the following knobs:

- Number of patches to split the input image $p$, choosing from $\{1, 2, 3, 4\}$ according to the input image resolution. The image will be split into $p \times p$ patches.
- Number of layers to run patch-based inference $n, n < N$, where $N$ is the total number of layers. The rest of the network will be run with per-layer inference.

**Training.** We follow the training protocol in [9] for super network training. The training dataset is randomly split into a sub-training set and validation set. The validation set size is 10,000 for ImageNet [3] and 5,000 for other datasets. We first train the largest network in the search space on the sub-training set using SGD with batch size 1024, initial learning rate 0.2, weight decay 4e-5, and a cosine learning rate decay. The training epochs is 150 for ImageNet [3] and 30 for VWW [2]. Afterward, we sort the channels according to their importance (we used L-1 norm for importance estimation [5]). Then we initialize the super network with the weights and then perform super network training using the same hyper-parameters for twice the epochs. For each iteration, 4 random architectures are sampled, and the gradients are averaged to train the network.

After getting the sub-network architecture from the evolutionary search, we fine-tuned the networks using 1/10 of the initial learning rate for 10 epochs.

**Validation.** To prevent over-fitting the real validation set, we evaluate the performance of each sub-network on the split validation set. The weights are taken from the super network using indexing. We re-calibrate the batch normalization statistics using 20 batches of data with a batch size 64.

**Evolutionary search.** We used evolutionary search to find the best sub-network architecture under certain constraints. We use a population size of 100. We randomly sample 100 sub-networks satisfying the constraints to form the first generation of population. For each iteration, we only keep the top-20 candidates with the highest accuracy. Then we perform crossover to generate 50 new candidates and mutation to generate another 50, forming a new generation. The mutation rate is 0.1. We repeat the process for 30 iterations and choose the sub-network with the highest accuracy on the split validation set.

**Quantization.** We perform int8 quantization following the format in [8]. To reduce the accuracy loss from quantization, we perform quantization-aware training for 10 epochs.

## C  Memory Distributions of Efficient Models

We further provide the memory distributions of three efficient models: MnasNet [12], FBNet [14], and MCUNet-320kB [9] in Figure S2. All the models have a highly imbalanced memory distribution, even for MCUNet, which is specialized for memory-constrained settings. The results demonstrate the generality of the imbalanced memory distribution phenomenon. Enabling patch-based inference can cut the peak memory usage of the models by 3.5-6.1×.

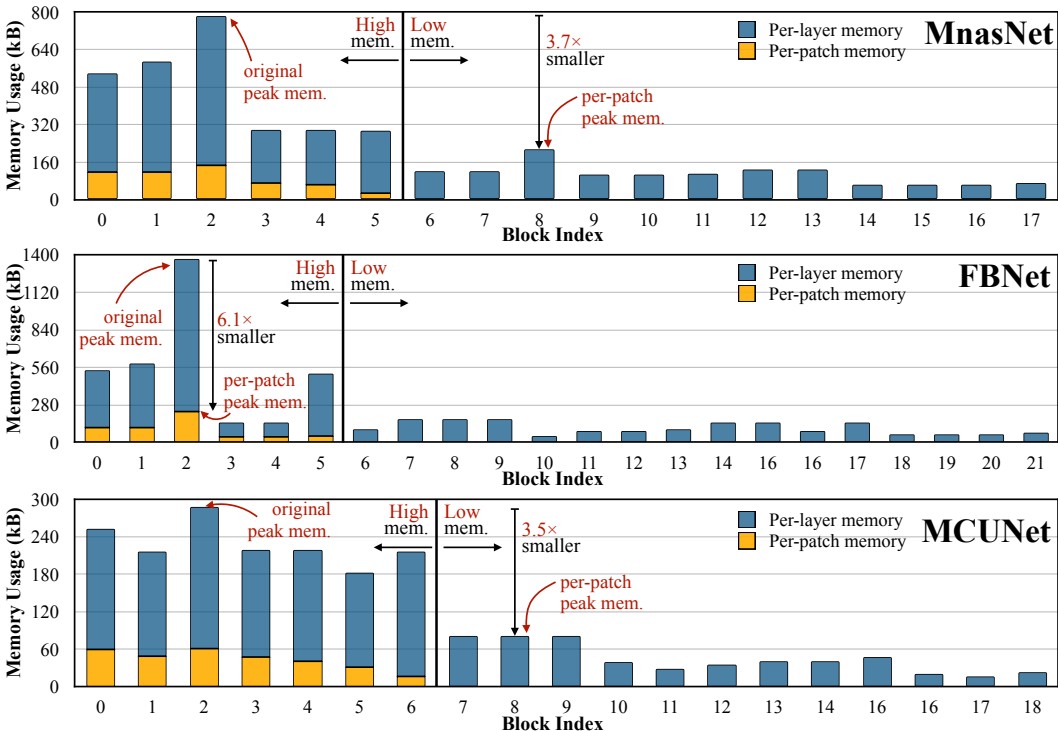

**Figure S2.** Memory distribution of MnasNet [12], FBNet [14], and MCUNet-320kB [9]. All the models have an imbalanced memory distribution. Enabling patch-based inference can reduce the peak memory by $3.5 - 6.1\times$.

# D    Ablation Study on Neural Architecture Search

Adding width multiplier $w$ and input resolution $r$ in the search space can greatly improve neural architecture search under tiny deep learning settings, because a flexible $r$ and $w$ allows us to globally *scale* the neural network to fit a tight resource budget. This is also mentioned as "search space optimization" in [9], where the authors proposed a two-step method that first chooses the optimal $w$ and $r$, and then performs neural architecture search under the given $w$ and $r$. Instead, we merge the two stages by directly adding $r$ and $w$ into the search space.

To show the advantage of our method, we conduct experiments on MobileNetV3 [6] space by extending it to support different $r$'s and $w$'s. We compared it with state-of-the-art methods under different computation budgets in Table S1. Our NAS method consistently outperforms existing techniques for tiny networks in terms of computation-accuracy trade-off. Existing techniques usually need a scaling method to scale down the searched network and fit different budgets. With the extended search space, all our models are derived from the *same* super network while obtaining the best accuracy. The accuracy improvement is more significant under a tiny computation setting ($\leq$25M). We also try supporting flexible $w$'s per block, which improves the accuracy for smaller computation budgets. Therefore, we enable flexible $w$'s by default in our experiments.

**Table S1.** Our NAS method outperforms existing state-of-the-art tiny networks in terms of computation-accuracy trade-off, especially under tiny computation settings (<50M). All our models are derived from *the same search space*, while obtaining the best accuracy at different budgets. For models with *, we re-measure the MACs and parameters using our profiler.

| Budget | Model | Setting | MACs | Weights | Top-1 | Top-5 |
|---|---|---|---|---|---|---|
| 100M MACs | MobileNetV1 0.5× (r=192) [7] | Manual+Scale | 110M | 1.3M | 61.7% | 83.6% |
| | MobileNetV2 0.75×(r=160) [11] | Manual+Scale | 107M | 2.6M | 66.4% | 87.3% |
| | MobileNetV3 Small 1.25× [6] | NAS+Scale | 91M | 3.6M | 70.4% | - |
| | EfficientNet-B$^{-2}$ [13, 4] | NAS+Scale | 98M | 3.0M | 70.5% | 89.5% |
| | TinyNet-C [4] * | NAS+Scale | 103M | 2.5M | 71.2% | 89.7% |
| | Ours (uniform $w$) | Joint Search | 98M | 4.2M | **72.3%** | **90.6%** |
| | Ours (flexible $w$) | Joint Search | 99M | 3.9M | **72.3%** | 90.5% |
| 50M MACs | MobileNetV2 0.35× [11] | Manual+Scale | 59M | 1.7M | 60.3% | 82.9% |
| | MnasNet-A1 0.35× [12] | NAS+Scale | 63M | 1.7M | 64.1% | 85.1% |
| | MnasNet-search1 [12] | NAS | 65M | 1.9M | 64.9% | - |
| | EfficientNet-B$^{-3}$ [13, 4] | NAS+Scale | 51M | 2.0M | 65.0% | 85.2% |
| | TinyNet-D [4] * | NAS+Scale | 53M | 2.3M | 67.0% | 87.1% |
| | MobileNetV3 Small 1.0× [6] | NAS | 56M | 2.5M | 67.4% | - |
| | Ours (uniform $w$) | Joint Search | 50M | 2.8M | 67.9% | 87.7% |
| | Ours (flexible $w$) | Joint Search | 50M | 3.5M | **68.8%** | **88.2%** |
| 25M MACs | MobileNetV2 0.35× (r=160) [11] | Manual+Scale | 30M | 1.7M | 55.7% | 79.1% |
| | MnasNet-A1 0.57× (r=128) [12] | NAS+Scale | 22M | 1.7M | 54.8% | 78.1% |
| | EfficientNet-B$^{-4}$ [13, 4] | NAS+Scale | 24M | 1.3M | 56.7% | 79.8% |
| | MobileNetV3 Small 0.5× [6] | NAS+Scale | 23M | 1.6M | 58.0% | - |
| | TinyNet-E [4] * | NAS+Scale | 25M | 2.0M | 59.9% | 81.1% |
| | Ours (uniform $w$) | Joint Search | 25M | 2.6M | 63.2% | 84.7% |
| | Ours (flexble $w$) | Joint Search | 25M | 3.2M | **63.9%** | **84.9%** |

# E  Qualitative Results of Face Detection

We provide the face detection results on WIDER FACE validation set with RNNPool-Face-Quant [10] and MCUNetV2-S. The quantitative results are shown in Table S2, where we follow [10] to calculate the peak memory. Our model has better mAP at $1.3\times$ smaller peak memory. The qualitative results are shown in Figure S3. Our model is more robust to poses and background false positives.

**Table S2.** MCUNetV2-S outperforms RNNPool-Face-Quant [10] on WIDER FACE at $1.3\times$ smaller peak memory.

| Method | MACs ↓ | Peak Memory ↓ (int8) | mAP ↑ | | | mAP (≤3 faces) ↑ | | |
|---|---|---|---|---|---|---|---|---|
| | | | Easy | Medium | Hard | Easy | Medium | Hard |
| RNNPool-Face-Quant [10] | 0.12G | 225kB ($1.3\times$) | 0.80 | 0.78 | 0.53 | 0.84 | 0.83 | 0.81 |
| MCUNetV2-S | **0.11G** | **168kB** ($1.0\times$) | **0.85** | **0.81** | **0.55** | **0.90** | **0.89** | **0.87** |

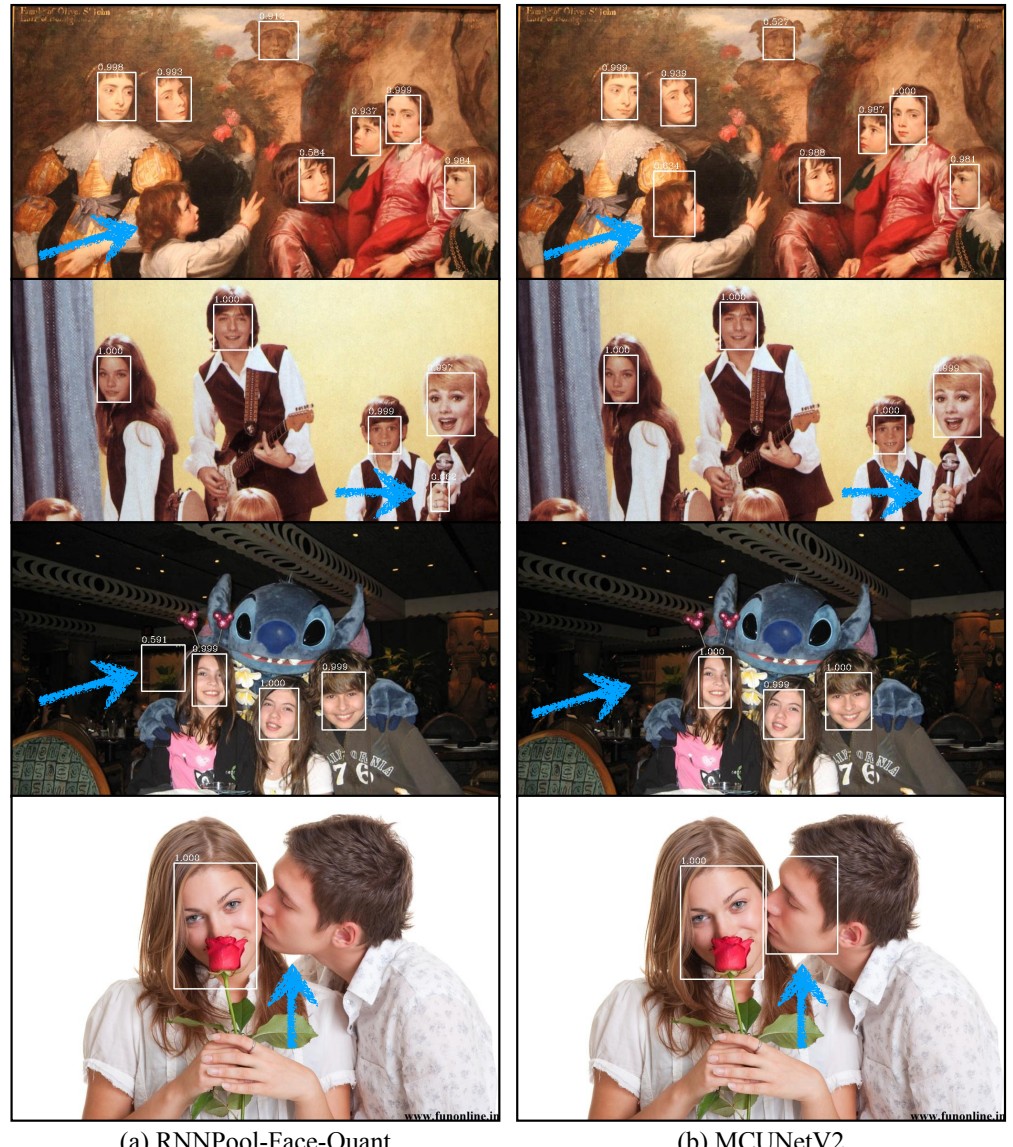

    (a) RNNPool-Face-Quant               (b) MCUNetV2

**Figure S3.** Qualitative results of face detection with RNNPool-Face-Quant [10] and MCUNetV2-S on WIDER FACE [15] validation set. Check the blue arrows: our model is more robust to poses and background false positives. The predictions are filtered with confidence threshold 0.5.