# OpenReview forum: "Memory-efficient Patch-based Inference for Tiny Deep Learning"
_NeurIPS.cc/2021/Conference — NeurIPS 2021 Poster_

### Official Review · Reviewer_VGan · 2021-07-13

**Rating:** 7
**Confidence:** 4

**Summary:**

This paper addresses the problem of large activations that quickly arises in devices with little memory, such as MCUs, when deploying larger networks. The main contribution proposes a relatively simple but effective approach to reducing the memory footprint of activations (by a large 4x-8x factor) while introducing some unavoidable extra compute (resulting in up to 15% MACs, which translate into  up to ~1.27x higher latency). The impact in terms of compute overhead of the proposed per-patch processing is alleviated by redistributing the receptive filed. Which means: reducing the filter WxH in early layers and use striding>1; then stack more blocks in deeper parts of the network where the memory peak of activations is not so high anymore. A NAS-based mechanism is introduced to jointly discover architectures and receptive filed redistribution strategies.


**Limitations And Societal Impact:**

The main limitation of the patch-based inference is well covered and addressed with the receptive field re-distribution analysis (first with MobileNetV2) and, later introduced in the NAS stage.

**Main Review:**


Large activations is arguably the main problem that’s preventing the deployment of more sophisticated ML-based applications to very constrained devices such as MCUs. This challenge is even worse in other domains such as graphs. This paper is very well presented and touches upon a known problem that has mainly been addressed from a neural architectural point of view but often only indirectly. The results are very compelling, specially in the case of object detection which, as the authors state, it is more sensible to reduced input or feature map resolutions. The list of references is complete, including recent works.

To revise/improve/expand:
- From Figure 3 and the first intuitive explanation given in the introduction and later in 3.1 in a bit more detail, I’m still not 100% sure how this is really implemented in practice. I would advice adding an \algorithm{} to describe how this is done.
- Could the authors elaborate on why/how PatchNN is 1.8x faster than MCUNet for the same ~92% on VWW (Figure6.b) , specially given the unavoidable extra compute needed for per-batch executions. Could the authors share details on the PatchNN architecture?
- Could the authors elaborate on what sorts of optimisations (e.g. im2row/col, SIMD etc) will be hard to make use of when relying on the proposed patch-based inference ?
- Minor: Line 51 says “Figure 3” but I believe the authors referred instead to “Figure 1”.

**Time Spent Reviewing:**

4

---

> ### Author Response · Authors · 2021-08-09
> **Author Response**
>
> **[1. Algorithm describing patch-based inference in Figure 3.]**
>
> Below we provide a pseudo-code describing the patch-based inference in Figure 3. Please find a **highlighted version** of the code in the [anonymous link](https://drive.google.com/file/d/1QuOaK6E6W5lZoDFTJa91UW39_4rfglzY/view?usp=sharing) for better readability.
>
>
> ```python
> def patch_based_inference(net, x, n_patch):
>    """
>    net: the network for inference
>    x: the input image
>    n_patch: number of patches to split the image for patch-based inference
>    """
>    net_patch, net_layer = split_net(net)  # split the network into per-patch stage and per-layer stage, net_patch(net_layer(x)) = net(x)
>    mid_feat = []  # to cache the intermediate outputs for each patch; the total size is still small
>    for i_patch in range(n_patch):
>        cur_patch = extract_patch(x, i_patch)  # extract the current patch from the input (much smaller than x)
>        mid_feat.append(net_patch(cur_patch))
>    mid_feat = combine_patch(mid_feat)  # combine patches into spatial dimension, e.g., combine 16 patches into 4x4 (h x w) spatial-wise
>    out = net_layer(mid_feat)  # per-layer inference
>    return out  # same as net(x)
> ```
>
> **[2. Why PatchNN is 1.8x faster than MCUNet for the same accuracy on VWW.]**
>
> Patch-based inference allows us to use a larger architecture design space given the same tight memory constraints (e.g., we are able to use a higher input resolution compared to per-layer inference). Therefore, we can design networks that have higher accuracy at the lower computation compared to MCUNet, leading to faster inference speed. On the other hand, the extra computation overhead for patch-based inference is negligible after joint optimization.
>
> We provide the visualization of the network architectures on VWW for MCUNet and PatchNN in the [anonymous link](https://drive.google.com/file/d/1O-lhR_s4vcvZwO1DzlPp7bNzcwoWoXy1/view?usp=sharing).
>
> (**Legend**: the color represents the kernel size, the height of each block represents the expansion ratio. The name is `MB{expansion ratio} {kernel_size}x{kernel_size}`). The dashed border means the block has stride=2. `{}x{}` in the bottom denotes the resolution of feature maps.)
>
> We can find that patch-based inference allows PatchNN to use a larger input resolution and a smaller model, achieving higher accuracy at a lower latency and peak memory. The same model could not run under 320kB SRAM with per-layer inference due to OOM. On the other hand, due to the memory limit, MCUNet has to increase the depth to improve performance, which is not the optimal tradeoff.
>
> The statistics of the two models are provided in the following table.
>
> |           | MCUNet | PatchNN   |
> | --------- | ------ | --------- |
> | MACs      | 57M    | **40M**   |
> | Latency   | 896ms  | **510ms** |
> | Accuracy  | 92%    | **93.6%** |
> | Peak SRAM | 319kB  | **104kB** |
>
> **[3. What sorts of optimizations will be hard to use under patch-based inference.]**
>
> We can still use the same optimization techniques (e.g., im2row/col, SIMD) as in per-layer inference. The current SIMD optimizations parallelize the _channel_ dimension (as in TF-Lite Micro, CMSIS-NN, TinyEngine), which is not affected by the _spatial_ dimension. Some techniques do rely on spatial parallelism; nonetheless, the single patch resolution is still large enough to provide enough parallelism (the smallest patch size is 7x7 in our study; mostly it is larger than 14x14). Multiplied with the parallelism in the channel dimension, the parallelism is more than enough on tiny hardware like MCUs with limited computation resources (only 2 MACs per cycle).
>
> **[4.Typos.]**
>
> Thanks. We will correct the typos in the final version.

---

> > ### Comment · Reviewer_VGan · 2021-08-23
> > **This is a good paper.**
> >
> > I would like to thank the authors for addressing the points I mentioned when reviewing this paper. I stand by my original score of 7

---

### Official Review · Reviewer_8aP4 · 2021-07-15

**Rating:** 7
**Confidence:** 4

**Summary:**

The main problem the paper tries to tackle is reducing the peak memory footprint of the neural networks by deviating from the usual layer-by-layer execution. The main intuition seems to be very similar to tiling combined with layer fusion-like ideas. However, the main spark of the paper seems to be leveraging this dimension in the context of Neural Architecture Search, which seems to be novel.

**Main Review:**

I believe the paper tackles a very important topic or reducing peak memory footprint of neural network execution. Especially, considering that many modern frameworks rely only on layer-by-layer computation based on the fact that it is mainly designed for less memory-constrained scenarios, what the paper proposes as patch-based is a good direction for MCUs.

I believe the main insight in the paper that should be highlighted is leveraging this as a new dimension for NAS. Overall, it seems that the patch-based nature allows smaller footprint and NAS acts as a good lubricant to leverage this aspect more naturally.

Single request that I have is the visualization of the output network architecture. I believe this would help the readers gain better insight about the work.

As a side note, I think the paper is missing a citation regarding reordering the operator execution to enable execution on MCUs have been extensively studies in [1.

[1] Ahn, B. H., Lee, J., Lin, J. M., Cheng, H. P., Hou, J., & Esmaeilzadeh, H. (2020). Ordering chaos: Memory-aware scheduling of irregularly wired neural networks for edge devices. arXiv preprint arXiv:2003.02369.


**Time Spent Reviewing:**

1-2

---

> ### Author Response · Authors · 2021-08-09
> **Author Response**
>
> Thanks so much for the constructive feedback and detailed comments.
>
> **[1. Visualization of output network architecture and insights.]**
>
> Thanks for the suggestion. We have provided the architecture comparison of MobileNetV2 and redistributed MobileNetV2 in supplementary D. Here we visualize the architecture of a PatchNN model on the VWW dataset in the [anonymous link](https://drive.google.com/file/d/1OROd_EEYErLaloc1na0_agLFHDuVLYjD/view?usp=sharing).
>
> (**Legend**: the color represents the kernel size, the height of each block represents the expansion ratio. The name is `MB{expansion ratio} {kernel_size}x{kernel_size}`). The dashed border means the block has stride=2. `{}x{}` in the bottom denotes the resolution of feature maps.)
>
>
> We have several observations:
>
> - The kernel size in the per-patch inference stage is small (always 3x3 before downsampling with stride=2) to reduce the spatial overlapping and computation overhead.
> - The expansion of the middle stage (early in per-layer stage) is small to further reduce the peak memory (larger expansion ratio leads to larger feature maps for depthwise convolution).
> - Large expansion ratio and large kernel size usually do not appear together to reduce the computational cost: if the expansion ratio is large (like 6), the kernel size is small (3x3 or 5x5); if the kernel size is large (7x7), the expansion ratio is small (3 or 4).
> - The input resolution is larger on some resolution-sensitive datasets like VWW compared to MCUNet, since per-layer inference cannot fit a large input resolution.
>
> The architecture is designed automatically with joint NAS without human prior, showing the effectiveness of joint NAS.
>
> **[2. Missing citation.]**
>
> Sorry for the missing citation. We will definitely add it to the updated version.

---

> > ### Comment · Reviewer_8aP4 · 2021-08-21
> > **I stand by my original score**
> >
> > I would like to thank the authors for providing the visualization.
> > I believe this paper serves nice contribution in the on-device AI research. While the overall method may look simple, I believe the idea provides reasonable gains in terms of efficiency which is imperative to better enable on-device AI. Therefore, I think this paper has great practical value.

---

### Official Review · Reviewer_rwM3 · 2021-07-16

**Rating:** 7
**Confidence:** 4

**Summary:**

Working memory utilization for most CNN based models typically is more than what can be afforded on Tiny devices. Thus, it is often quite difficult to extract reasonable performance for visual tasks on tiny devices.

Authors attack this problem by a) introducing a patch based inference method that improves peak memory utilization at the cost of compute and b) introducing a redistribution process that reduces the overall compute while maintaining/improving performance.

**Limitations And Societal Impact:**

Yes.

**Main Review:**

Working memory utilization for most CNN based models typically is more than what can be afforded on Tiny devices. Thus, it is often quite difficult to extract reasonable performance for visual tasks on tiny devices. The high memory utilization is a result of storing activation between layers in CNN inference. In a typical layer by layer inference implementation, one has to store the intermittent activation in memory while the next set of activation are computed. This means that for large layers or layers with a lot of filters, there is a high working memory requirement.

To attack this problem, authors propose a patch based inference method -- essentially deviating from the layer-by-layer evaluation to a more 'patch-by-patch' evaluation method. This approach critically depends on the fact that later layers in CNN architectures are typically smaller in their activation sizes, and one can compute each entry of these smaller activations while by streaming in its respective receptive field --- we only need to store the smaller activation of this later layer in memory.

Of course such a patch based inference often comes with the cost of extra computation. To counteract this, the authors propose modifying the network architecture to be computation heavy on the later layers. Authors propose a NAS approach to guide this modification (and show certain advantages of this approach like a larger design space). Authors evaluate their method experimentally and validate their claims on tiny-ml benchmarks. The experimental results on existing datasets look very promising, with good/better performance metrics than existing models at marginally more computational costs/latency while hitting SRAM requirements. Similar results are also reported for Tiny image classification with PatchNN offering a better SRAM-accuracy and latency-accuracy tradeoffs.

Overall  I feel that the work attacks an important problem and presents a very viable, conceptually simple solution with good empirical performance.

**Time Spent Reviewing:**

4

---

> ### Author Response · Authors · 2021-08-09
> **Author Response**
>
> Thanks so much for the supportive comments. We are glad that the reviewer finds our addressed problem important. Please let us know if you have any questions.

---

> > ### Comment · Reviewer_rwM3 · 2021-08-23
> > **Retain my score**
> >
> > Thank you so much for the detailed response across all reviews. After going through them, I am retaining my score and believe this work will be a valuable contribution at NeurIPS.

---

### Official Review · Reviewer_bZRz · 2021-07-27

**Rating:** 6
**Confidence:** 5

**Summary:**

Instead of the conventional layer-by-layer execution, this paper proposes a patch-by-patch scheduler to reduce the peak memory required by a CNN for inference on MCU.
The author also proposes a network redistribution method, which finetunes the network architecture (CONV kernel size), to mitigate the computation overhead caused by their patch-by-patch execution. The results show that, with a certain computation overhead, the proposed methods can effectively reduce the inference peak memory while not losing accuracy.

**Limitations And Societal Impact:**

Yes

**Main Review:**

The MCU devices has drawn great attention recently, and this paper targets memory-efficient inference on MCU devices. The proposed method can effectively reduce the peak memory-bound during the NN inference. Since the peak memory is reduced, it allows the MCU device to accommodate larger NN or using larger input size to improve the accuracy. This provides the proposed method a clear advantage when compared to the baseline works. This paper is well-written and easy to follow.

However, the search method used in this paper also follows the previous works with some additional search space.

The author shows the classification results on large dataset (e.g., ImageNet), which is great. But for the object detection results, the author shows the results on VOC dataset, which is considered an easy dataset. The author proposes the network redistribution that will change the receptive field of the well-designed network structure. It is unknown whether it may degrade the performance on larger datasets e.g., MS-COCO. Because the object detection tasks may be more sensitive to the receptive field changes.
It seems that the patch-by-patch and network redistribution techniques quite rely on the proposed joint NAS to reduce the computation overhead and maintain accuracy. So, it would be good to know that whether the proposed joint NAS is still valid and affordable on larger datasets for object detection tasks such as MS-COCO.

In the introduction section, the author claims that the pruning, quantization, and NAS methods ``focus on reducing the number of parameters and FLOPs, which cannot address the memory bottleneck’’. I think this is not correct since those model compression techniques can reduce the memory cost for both model weights and activations in an order of magnitude manner.

For branches network structure, the author mentioned that the memory requirement is calculated by summing all the memory required by each branch together. But it may overestimate the peak memory, because same branches may be able to share the input or output activations.

Minor issues:
A typo in line 65 “On ImageNet”
Since the inference process only uses 1 picture at a time, it would be good to know the percentage of memory cost by the weights.


**Time Spent Reviewing:**

5 hours

---

> ### Author Response · Authors · 2021-08-09
> **Author Response**
>
> Thanks so much for the constructive feedback and detailed comments.
>
> **[1. Contribution highlight: patch-based inference schedule, not search method.]**
>
> We only leverage the search technique to automatically optimize the network and inference system and ease human labor. Other search techniques (e.g., reinforcement learning, Bayesian optimization) are also feasible in our cases.
>
> The main contributions of our paper are:
>
> - Systematically analyze the memory usage pattern of efficient CNN designs and point out the critical issue of imbalanced memory distribution for the TinyML setting.
> - Propose a patch-based inference scheduling to significantly reduce the peak memory for inference.
> - Propose network redistribution technique to reduce the computation overhead from overlapping patches.
> - Leverage automated search techniques to jointly optimize the neural architecture and inference scheduling.
>
>
> **[2. Network redistribution will not degrade the performance on larger object detection datasets like MS-COCO.]**
>
> Thanks for the suggestion. Firstly, the network redistribution technique only _shifts_ parts of the receptive field from the early network stage to the later stage, but not _reduces_ the overall receptive field. The input features to the detection head still have the same receptive field compared to the original network. With the technique, we can further accommodate a larger input resolution given the same memory constraint, which is critical to object detection.
>
> We validate the object detection performance on the MS-COCO dataset. We compare the object detection performance of the MobileNetV2 backbone (MbV2) with YOLO-V3 and its redistributed version (MbV2-RD) in the following table (extension of Table 1 in the original paper). MbV2-RD has similar or slightly better performance on both VOC and COCO datasets.
>
> |         | VOC mAP | COCO mAP | COCO AP50 |
> | ------- | ------- | -------- | ------------ |
> | MbV2    | 75.4%   | 24.8%    | 44.1%        |
> | MbV2-RD | 75.7%   | 25.0%    | 44.3%        |
>
> We further compare the object detection performance fitting MCUs of 512kB SRAM limit (extension of Table 3 of the original paper). The improvement on COCO is consistent and significant.
>
> | Model       | Peak SRAM | VOC mAP | COCO mAP | COCO AP50 |
> | ----------- | --------- | ------- | -------- | ------------ |
> | MCUNet [26] | 466kB     | 51.4%   | 13.1%    | 26.1%        |
> | PatchNN     | 438kB     | 68.3%   | 16.7%    | 32.8%        |
>
> **[3. Joint NAS is valid and affordable on larger detection datasets like MS-COCO.]**
>
> For joint NAS, we used a one-shot NAS method based on once-for-all networks [4]. It only requires training the super network once at roughly 2x of the cost compared to normal network training. For the search part, we need to sample a valid neural architecture and inference scheduling under constraints (which can be verified instantly) and measure the performance on a hold-out validation set, consuming a modest computation cost.
>
> We have performed the joint NAS on the large-scale image classification dataset ImageNet in the paper. The COCO dataset consumes the same level of training cost compared to ImageNet. Please see the COCO results in the table above which demonstrates the feasibility.
>
> **[4. Claim that pruning, quantization, and NAS cannot address the memory bottleneck.]**
>
> Sorry for the confusion. The claim is indeed improper. Model compression techniques can reduce the memory cost. But traditional mobile architecture designs target reducing model size or computation, not the inference peak memory (which is more related to activation size). For example, compared to ResNet-18, MobileNetV2 has 4.6x smaller model size and 6x smaller computation, but the peak memory is 1.8x larger [26]; while PatchNN reduces both computation and peak memory. We will correct the sentence in the final version.
>
> The model compression techniques and our proposed method are orthogonal. They can be combined to further improve memory efficiency.
>
> **[5. Overestimate the peak memory in a multi-branch setting.]**
>
> For multi-branch cases, if some branches share the same input or output activation, we only count the shared activation once to ensure no repeated calculation. We will add the details in the final version.
>
> **[6. Percentage of memory cost by the weights.]**
>
> All the analysis in this paper is based on batch size 1, and the activation still dominates the peak memory.
>
> During the model inference on MCUs, the weights of each layer can be partially fetched from the Flash and released after computation, thus it will not become the memory bottleneck. For the backbone of PatchNN-M4 on ImageNet (Table 2), the analytic peak memory is 1089kB (per-layer inference) and 196kB (per-patch inference). The biggest weight tensor for each layer is 60kB. We can see that the activation size is the major bottleneck, which could be addressed by patch-based inference.
>
> **[7. Typos.]**
>
> Thanks. We will correct the typos in the final version.
>
>
> **Reference**
>
> [4] Han Cai, Chuang Gan, Tianzhe Wang, Zhekai Zhang, and Song Han. Once for All: Train One Network and Specialize it for Efficient Deployment. In ICLR, 2020
>
> [26] Ji Lin, Wei-Ming Chen, Yujun Lin, John Cohn, Chuang Gan, and Song Han. MCUNet: Tiny Deep Learning on IoT Devices. In NeurIPS, 2020.

---

### Author Response · Authors · 2021-08-09
**General Response**

We sincerely appreciate all reviewers’ time and efforts in reviewing our paper and the constructive feedback. We genuinely appreciate all reviewers’ positive initial impressions. In addition to the pointwise response below, here we summarize our updates:

- **[Extra experiments on MS-COCO]** We further provide the results of object detection on a larger-scale dataset MS-COCO. The consistent performance gain demonstrates the effectiveness and scalability of our method.
- **[Visualization of PatchNN architecture]** We provide the visualization of some PatchNN model architectures and summarize some design insights automatically discovered by the joint NAS algorithm.
- **[Pseudo code of patch-based inference]** We provide a pseudo code to further clarify the process of patch-based inference.

We hope our response below could address all reviewers’ concerns. We thank all reviewers' efforts again.

---

### Decision · Program_Chairs · 2021-09-27

**Decision:**

Accept (Poster)

**Comment:**

The authors tackle an important and salient problem: inference on microcontrollers for DL, where memory is very constrained. The proposed scheme, patch by patch inference, works well, the reviewers were happy with the paper and the method itself, therefore I recommend acceptance.